# CONTEXT DEPENDENT MODULATION OF ACTIVATION FUNCTION

## ABSTRACT

We propose a modification to traditional Artificial Neural Networks (ANNs), which provides the ANNs with new aptitudes motivated by biological neurons. Biological neurons work far beyond linearly summing up synaptic inputs and then transforming the integrated information. A biological neuron change firing modes accordingly to peripheral factors (e.g., neuromodulators) as well as intrinsic ones. Our modification connects a new type of ANN nodes, which mimic the function of biological neuromodulators and are termed modulators, to enable other traditional ANN nodes to adjust their activation sensitivities in run-time based on their input patterns. In this manner, we enable the slope of the activation function to be context dependent. This modification produces statistically significant improvements in comparison with traditional ANN nodes in the context of Convolutional Neural Networks and Long Short-Term Memory networks.

## 1 INTRODUCTION

Artificial neural networks (ANNs), such as convolutional neural networks (CNNs) (LeCun et al., 1998) and long short-term memory (LSTM) cells (Hochreiter & Schmidhuber, 1997), have incredible capabilities and are applied in a variety of applications including computer vision, natural language analysis, and speech recognition among others. Historically, the development of ANNs (e.g., network architectures and learning algorithms) has benefited significantly from collaborations with Psych-Neuro communities (Churchland & Sejnowski, 1988; Hebb, 1949; Hinton et al., 1984; Hopfield, 1982; McCulloch & Pitts, 1943; Turing, 1950; Hassabis et al., 2017; Elman, 1990; Hopfield & Tank, 1986; Jordan, 1997; Hassabis et al., 2017). The information processing capabilities of traditional ANN nodes are rather rigid when compared to the plasticity of real neurons. A typical traditional ANN node linearly integrate its input signals and run the integration through a transformation called an activation function, which simply takes in a scalar value and outputs another. Of the most popular Activation Functions are *sigmoid* (Mikolov et al., 2010), *tanh* (Kalman & Kwasny, 1992) and *ReLU* (Nair & Hinton, 2010).

Researchers have shown that it could be beneficial to deploy layer-/node- specific activation functions in a deep ANN (Chen & Chang, 1996; Solazzi & Uncini, 2000; Goh & Mandic, 2003; He et al., 2015; Agostinelli et al., 2014). However, each ANN node is traditionally stuck with a fixed activation function once trained. Therefore, the same input integration will always produce the same output. This fails to replicate the amazing capability of individual biological neurons to conduct complex nonlinear mappings from inputs to outputs (Antic et al., 2010; Hassabis et al., 2017; Marblestone et al., 2016). In this study, we propose one new modification to ANN architectures by adding a new type of node, termed modulators, to modulate the activation sensitivity of the ANN nodes targeted by modulators (see Figures 1-3 for examples). In one possible setting, a modulator and its target ANN nodes share the same inputs. The modulator maps the input into a modulation signal, which is fed into each target node. Each target node multiples its input integration by the modulator signal prior to transformation by its traditional activation function. Examples of neuronal principles that may be captured by our new modification include intrinsic excitability, diverse firing modes, type 1 and type 2 forms of firing rate integration, activity dependent facilitation and depression and, most notably, neuromodulation (Marder et al., 1996; Sherman, 2001; Ward, 2003; Ringrose & Paro, 2004).

Our modulator is relevant to the attention mechanism (Larochelle & Hinton, 2010; Mnih et al., 2014), which dynamically restricts information pathways and has been found to be very useful in practice. Attention mechanisms apply the attention weights, which are calculated in run-time, to the outputs of ANN nodes or LSTM cells. Notably, the gating mechanism in a Simple LSTM cell can also be viewed as a dynamical information modifier. A gate takes the input of the LSTM cell and outputs gating signals for filtering the outputs of its target ANN nodes in the same LSTM cell. A similar gating mechanism was proposed in the Gated Linear Unit (Dauphin et al., 2016) for CNNs. Different from the attention and gating mechanisms, which are applied to the outputs of the target nodes, our modulation mechanism adjusts the sensitivities of the target ANN nodes in run-time by changing the slopes of the corresponding activation functions. Hence, the modulator can also be used as a complement to the attention and gate mechanisms.

Below we will explain our modulator mechanism in detail. Experimentation shows that the modulation mechanism can help achieve better test stability and higher test performance using easy to implement and significantly simpler models. Finally, we conclude the paper with discussions on the relevance to the properties of actual neurons.

## 2 METHODS

We designed two modulation mechanisms, one for CNNs and the other for LSTMs. In modulating CNNs, our modulator (see Figure 1) is a layer-specific one that is best compared to the biological phenomenon of neuromodulation. Each CNN layer before activation has one modulator, which shares the input $\vec{x}$ with other CNN nodes in the same layer (Figure 1Left). The modulator (Figure 1Right) of the $l^{th}$ CNN layer calculates a scalar modulation signal as $s_l = \tau_l(\vec{w}_l^T \vec{x})$, where $\tau_l(\cdot)$ is the activation function of the $l^{th}$ modulator, and feeds $s_l$ to every other CNN node in the same layer. The $k^{th}$ modulated CNN node in the $l^{th}$ layer linearly integrates its inputs as a traditional ANN nodes $v_{l,k} = \vec{w}_{l,k}^T \vec{x}$ and modulates the integration to get $u_{l,k} = s_l \cdot v_{l,k}$ prior to its traditional activation step $\varphi_{l,k}(\cdot)$. The final output is $o_{l,k} = \varphi_{l,k}(\tau_l(\vec{w}_l^T \vec{x}) \cdot \vec{w}_{l,k}^T \vec{x})$. The above modulation mechanism is slightly modified to expand Densely Connected CNNs (Iandola et al., 2014)(see Figure 2). A modulator is added to each dense block layer to modulate the outputs of its convolution nodes. Given a specific input, the modulator outputs a scalar modulation signal that is multiplied to the scalar outputs of the target convolution nodes in the same layer.

In addition to the Cellgate, there are three modifying gates (Forget, Input, and Output) in a traditional LSTM cell. Each gate is a full layer of ANN nodes. Each of ANN node in a gate uses $sigmoid$ to transform the integration of the input into regulation signals. The traditional LSTM cell transforms the input integration to an intermediate output (i.e., $\tilde{C}_t$ in Figure 3). The Forget gate regulates what is removed from the old cell state (i.e., $_{t-1}$ in Figure 3), and the Input gate what in $\tilde{C}_t$ is added to obtain the new cell state (i.e., $_t$). The new cell state is transformed and then regulated by the output gate to become part of the input of the next time point. In modulating LSTM (see Figure 3), for the purpose of easier implementation, we create a new "modulation gate" (the round dash rectangle in Figure 3) for node-specific sensitivity-adjustment which is most analogous to neuronal facilitation and depression. Different from a conventional LSTM that calculates $\tilde{C}_t = \varphi(W_c[\vec{x}_t, \vec{h}_{t-1}])$, a modulated LSTM calculates $\tilde{C}_t = \varphi(\tau(W_M[\vec{x}_t, \vec{h}_{t-1}]) \cdot (W_c[\vec{x}_t, \vec{h}_{t-1}]))$.

In the above designs, both a multi-layer CNN and single-layer LSTM had multiple modulator nodes within each model. A generalization to the above designs is to allow a modulator to take the outputs from other CNN layers or those of the LSTM cell at other time points as the inputs.

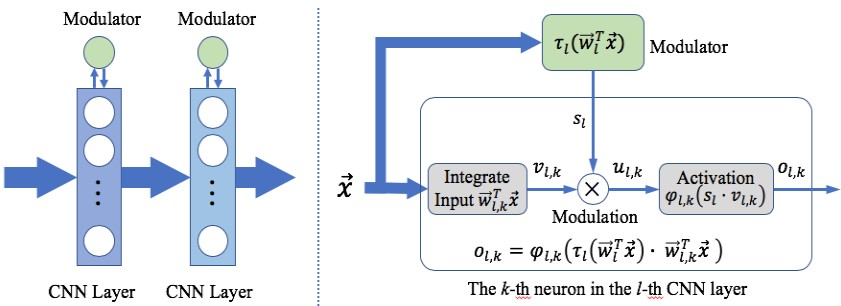

Figure 1: Modulator expansion to CNN. **Left**: Each CNN layer before activation layer has a modulator, whose input $\vec{x}$ is the same to that of other CNN nodes in the same layer. **Right**: The modulator maps the input into a scalar modulation signal, which is fed to every other CNN node in the same layer. Every modulated node multiples the integration of its inputs by the modulation signal prior to traditional activation.

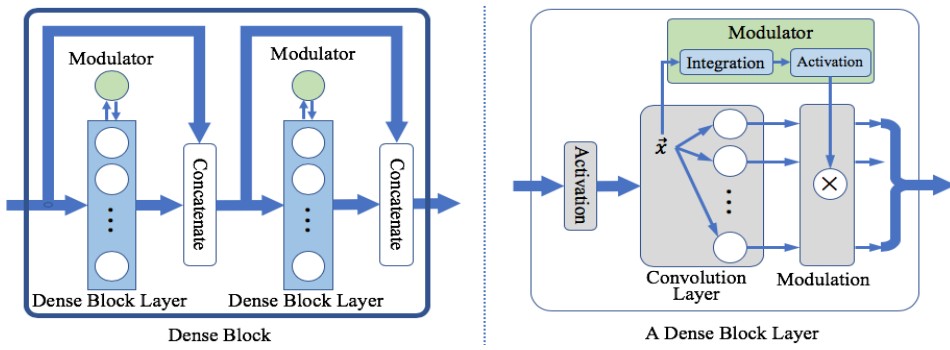

Figure 2: Modulator expansion to a dense block in a DenseNet. **Left**: A dense block example that contains two dense block layers. **Right**: A dense block layer transforms its input element-wisely before convolution. Hence, the modulation applied to the previous dense block layer affects the transformation activity of the next dense block layer. Modulation is applied to the output of each node in the target convolution layer. We ignore other steps (e.g, batch normalization and drop-out) in a dense block layer for visualization purpose.

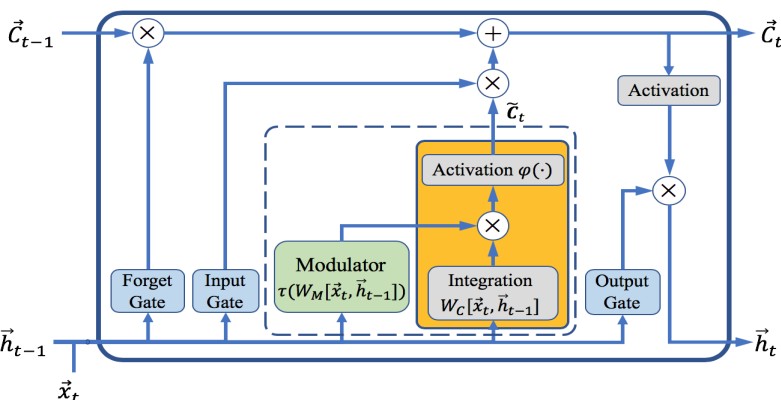

Figure 3: Modulator expansion to a Simple LSTM cell. A Simple LSTM cell wraps the Input Integration and Activation steps in the orange box into one node. Our modification adds a modulator (the green box) that maps $[\vec{x}_t, \vec{h}_{t-1}]$, i.e., the concatenation of the input at time $t$ and the LSTM output at time $t-1$, into modulation signals that are used to adjust the sensitivity of transforming the integration of $[\vec{x}_t, \vec{h}_{t-1}]$.

## 3 EXPERIMENTAL RESULTS

### 3.1 MODULATED CNNS

In our experiments with CNNs, the activation functions of the traditional CNN nodes was ReLU, with our modulator nodes using a *sigmoid*. We tested six total settings: a vanilla CNN *vs* a modulated vanilla CNN, a vanilla DenseNet *vs* a modulated DenseNet, and a vanilla DenseNet-lite *vs* a modulated DenseNet-lite. The vanilla CNN has 2 convolution blocks, each of which contains two sequential convolution layers, a pooling layer, and a dropout layer. A fully connected layer of 512 nodes is appended at the very end of the model. The convolution layers in the first block have 32 filters with a size of 3x3 while the convolution layers in the second block have 64 filters with a size of 3x3. We apply a dropout of 0.25 to each block. The vanilla DenseNet used the structure (40 in depth and 12 in growth-rate) reported in the original DenseNet paper (Iandola et al., 2014) and a dropout of 0.5 is used in our experiment. The vanilla DenseNet-lite has a similar structure to the vanilla DenseNet, however, uses a smaller growth-rate of 10 instead of 12 in the original configuration, which results in 28% fewer parameters. The modulators are added to the vanilla CNN, the vanilla DenseNet, and the vanilla DenseNet-lite in the way described in Figures 1 and 2 to obtain their modulated versions, respectively. Table 1 summarizes the numbers of the parameters in the above models to indicate their complexities. The modulated networks have slightly more parameters than their vanilla versions do. All the experiments were run for 150 epochs on 4 NVIDIA Titan Xp GPUs with a mini-batch size of 128.

Table 1: Compare the numbers of parameters in the CNNs.

| Model | Total parameters |
|---|---|
| vanilla CNN | 1.25 M |
| modulated CNN | 1.25 M |
| vanilla DenseNet | 1.00 M |
| modulated DenseNet | 1.07 M |
| vanilla DenseNet-lite | 0.71 M |
| modulated DenseNet-lite | 0.77 M |

CIFAR-10 dataset (Krizhevsky & Hinton, 2009) was used in this experiment. CIFAR-10 consists of colored images at a resolution of 32x32 pixels. The training and test set are containing 50000 and 10000 images respectively. We held 20% of the training data for validation and applied data augmentation of shifting and mirroring on the training data. All the CNN models are trained using the Adam (Kingma & Ba, 2014) optimization method with a learning rate of 1e-3 and shrinks by a factor of 10 at 50% and 80% of the training progress.

As shown in Figure 4, the vanilla CNN model begins to overfit after 80 training epochs. Although the modulated CNN model is slightly more complex, it is less prone to overfitting and excels its vanilla counterpart by a large margin (see Table 2). Modulation also significantly helps DenseNets in training, validation, and test. The modulated DenseNet/DenseNet-lite models consistently outperform their vanilla counterparts by a noticeable margin (see Figures 5(a) and 5(b)) during training. The validation and test results of the modulated DenseNet/DenseNet-lite models are also better than those of their vanilla counterparts. It is not surprising that the vanilla DenseNet-lite model underperforms the vanilla DenseNet model. Interestingly, despite having 28% fewer parameters than the vanilla DenseNet model, the modulated DenseNet-lite model outperforms the vanilla DenseNet model (see the dash orange curve vs the solid blue curve in Figure 5(b) and Table 2).

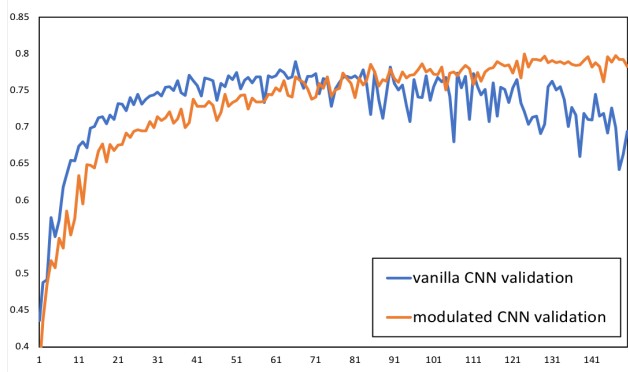

Figure 4: Modulation increases the robustness of CNNs. The x-axis is the validation epoch, and the y-axis is the top-1 accuracy. The vanilla CNN begins to show sign of overfitting after 80 epochs, while the modulated CNN keeps improving.

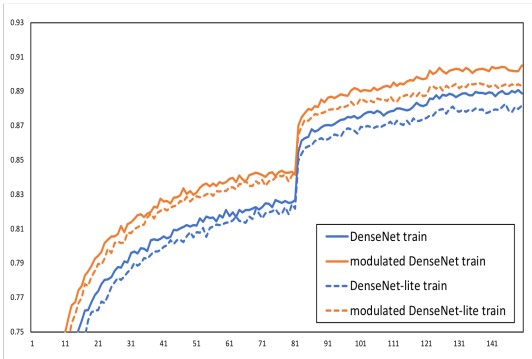

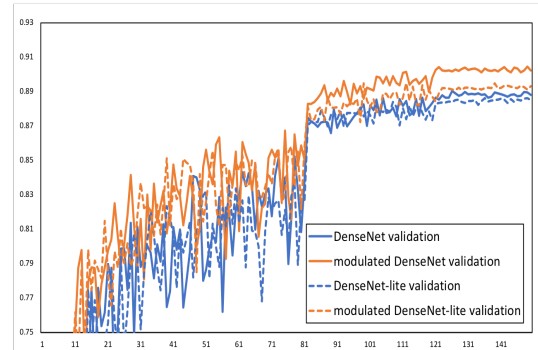

(a) Comparison of the training results of vanilla DenseNet, moduldate DenseNet, vanilla DenseNet-lite, and modulate DenseNet-lite. The x-axis is the training epoch, and the y-axis is the top-1 accuracy.

(b) Comparison of the validation results of vanilla DenseNet, modulate DenseNet, vanilla DenseNet-lite, and modulate DenseNet-lite. The x-axis is the validation epoch, and the y-axis is the top-1 accuracy.

Figure 5: Modulation improves the performance of DenseNets. Both training and validation results show that the modulated versions perform better than their vanilla versions. In addition, modulated DenseNet-lite clearly outperforms vanilla DenseNet even though the former has 28% less parameters than the latter.

Table 2: Test results (top-1 classification accuracy) of vanilla and modulated CNNs.

| Model | Training | Validation | Test |
|---|---|---|---|
| vanilla CNN | 0.649 | 0.694 | 0.690 |
| modulated CNN | 0.775 | 0.782 | 0.782 |
| vanilla DenseNet | 0.889 | 0.888 | 0.889 |
| modulated DenseNet | **0.905** | **0.902** | **0.902** |
| vanilla DenseNet-lite | 0.881 | 0.885 | 0.885 |
| modulated DenseNet-lite | **0.894** | **0.893** | **0.894** |

## 3.2 MODULATED LSTM

Two datasets were used in the LSTM experiments. The first one is the NAMES dataset (Sean, 2016), in which the goal is to take a name as a string and classify its ethnicity or country of origin. Approximately 10% of the data-set was reserved for testing. The second experiment used the SST2

data-set (Socher et al., 2013), which requires a trained model to classify whether a movie review is positive or negative based on the raw text in the review. The SST2 is identical to the SST1 with the exception of the neutral category removed (Socher et al., 2013), leaving only positive and negative reviews. About 20% of the data-set was reserved for testing.

Since modulators noticeably increase the parameters in a modulated LSTM, to perform fair comparisons, we create three versions of vanilla LSTMs (see Controls 1, 2, & 3 in Figure 6). Control 1 has an identical total LSTM cell size. Control 2 has the identical number of nodes per layer. Control 3 has an extra Input gate so that it has both an identical total number of nodes and identical nodes per layer. The numbers of parameters in the modulated LSTM and control LSTMs are listed in Table 3 for comparison.

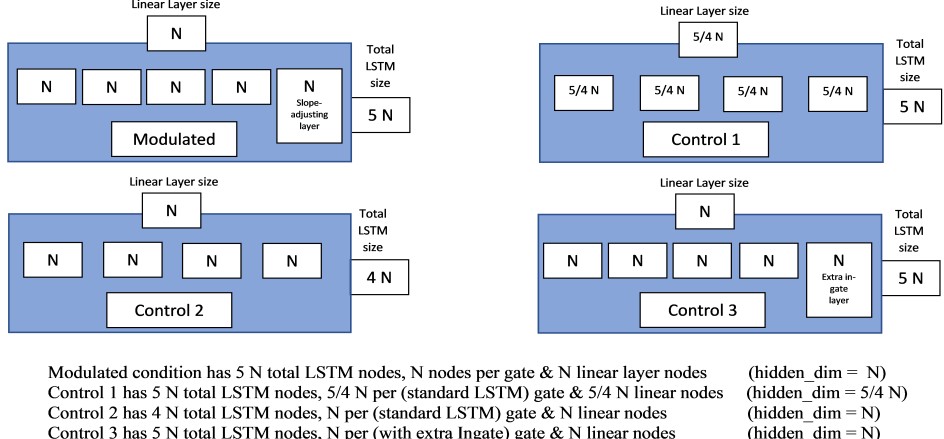

Modulated condition has 5 N total LSTM nodes, N nodes per gate & N linear layer nodes (hidden_dim = N)
Control 1 has 5 N total LSTM nodes, 5/4 N per (standard LSTM) gate & 5/4 N linear nodes (hidden_dim = 5/4 N)
Control 2 has 4 N total LSTM nodes, N per (standard LSTM) gate & N linear nodes (hidden_dim = N)
Control 3 has 5 N total LSTM nodes, N per (with extra Ingate) gate & N linear nodes (hidden_dim = N)

Figure 6: The configurations of the modulated LSTM and three control versions of the vanilla LSTMs.

The hyper-parameters for the first experiment were set as following: the hidden dimension was set to 32, batch size to 32, embedding dimension to 128, initial learning rate to .01, learning rate decay to 1e-4, an SGD optimizer was used, with dropout of 0.2 applied to the last hidden state of the LSTM and 100 epochs were collected. This condition was tested on the name categorization data-set. The number of parameters in this model ranged from 4.1 K to 6.4 K, depending on the condition. We repeated the experimental runs 30 times. Based on the simplicity of the data-set and the relative sparsity of parameters, this condition will be referred to as *Simple-LSTM*. As for the second experiment: the hidden dimension was set to 150, he batch size was set to 5, the embedding dimension was set to 300, the initial learning rate was set to 1e-3, there was no learning rate decay, an Adam optimizer was used with no dropout and 100 epochs were collected. The number of parameters in this model ranged from 57.6 K to 90 K, depending on the control setup. This experiment was repeated 100 times. Based on the complexity of the data-set and the relatively large amount of parameters, this condition will be referred to as *Advanced-LSTM*. In all experiments, the models were trained for 100 epochs.

Table 3: The number of parameters in a LSTM cell.

| Base Model | Modulated | Control 1 | Control 2 | Control 3 |
|---|---|---|---|---|
| Simple-LSTM | 5.1 K | 6.4 K | 4.1 K | 5.1 K |
| Advanced-LSTM | 72 K | 90 K | 57.6 K | 72 K |

Table 4: Modulation Performance Tests on LSTM

| Model Dataset | Modulator Activation | Performance Statistics | Modulated | Control 1 | Control 2 | Control 3 |
|---|---|---|---|---|---|---|
| Simple-LSTM *NAMES* | **Sigmoid** | mean | **0.77898** | 0.77490 | 0.77516 | 0.77602 |
| | | std | 0.00526 | 0.00565 | 0.00531 | 0.00650 |
| | | Hedge's G | - | 0.74778 | 0.72173 | 0.49924 |
| | | *p*-value | - | ***p<.006*** | ***p<.007*** | ***p<.06*** |
| Advanced-LSTM *SST2* | **Tanhshrink** | mean | **0.77482** | 0.73629 | 0.74327 | 0.73628 |
| | | std | 0.01150 | 0.04855 | 0.04687 | 0.04701 |
| | | Hedge's G | - | 1.09182 | 0.92435 | 1.12593 |
| | | *p*-value | - | ***p<.001*** | ***p<.001*** | ***p<.001*** |

Table 5: LSTM node Activation function visualizations

Effect of $Ingate_i * Tanh(Cellgate_i)$
(Control condition)

Effect of $Ingate_i * Tanh(Modulator_i * Cellgate_i)$
(Modulated condition)

**Simple-LSTM** (*sigmoid $\tau_l$*)

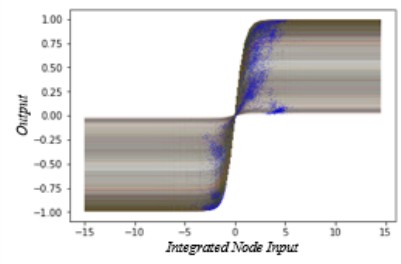 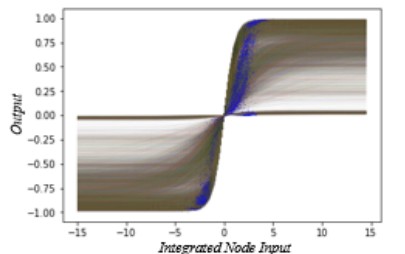

In this node, the Ingate produces diverse clusters of potential outputs for a given input. However, specific input-output relationships seem somewhat poorly defined, illustrated by the low density of the clusters.

The added source of variability from the adjustable slope appears to fine-tune the Activation clusters in a homeostatic manner. Four Activation clusters and three Activation Function modes are distinguishable.

**Advanced-LSTM** (*tanhshrink $\tau_l$*)

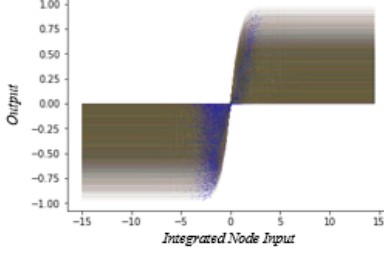 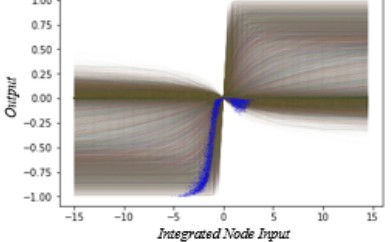

In this node, the Activation clusters produced by the Ingate appear to be rather undefined. This approach may be interpretable as explorative, since the two dimensional space is being examined, but this comes at the cost of having poorly-defined boundaries.

The boundaries of the input-output relationship are well-defined since the Activation clusters are denser. Due to the potential for negative slopes, this node has learned an input-modulator interaction that produces only negative outputs, akin to an inhibitory neuron.

**Note: Individual plot – Individual node** (*Blue dots - sampled Activations*)

We can observe from the results in Table 4 that, the mean test performance of both modulated LSTMs outperformed all three control groups and achieved the highest validation performance. Statistical significance varied between the two LSTM models. In the Vanilla-LSTM (n = 30), with $\tau_l(\cdot)$ set to $sigmoid$, statistical significance ranged between **p**<.06 (Control 3) and P<.001 (Control 2). In the Advanced-LSTM (n = 100), with $\tau_l(\cdot)$ set to $tanhshrink$, statistical significance was a consistently P<.001 in all conditions. In all cases, variance was lowest in the modulated condition. We further zoom in the activation data-flow and visualized the the effect of our modulation in Table 3.2. The control condition and modulated condition was compared side by side. On the left we can observe the impact of the Ingate on the amplitude of the *tanh* activation function, on the right we can observe our modulation adjust the slope as well. Each input generates a context dependent activation as shown in continuous lines and specific activations are represented by the blue dots which corresponded to a point on a specific line. Our modulation modification provides new aptitudes for the model to learn, generalize and appears to add a stabilizing feature to the dynamic input-output relationship.

## 4    CONCLUSION

We propose a modulation mechanism addition to traditional ANNs so that the shape of the activation function can be context dependent. Experimental results show that the modulated models consistently outperform their original versions. Our experiment also implied adding modulator can reduce overfitting. We demonstrated even with fewer parameters, the modulated model can still perform on par with it vanilla version of a bigger size. This modulation idea can also be expanded to other setting, such as, different modulator activation or different structure inside the modulator.

## 5    DISCUSSION

It was frequently observed in preliminary testing that arbitrarily increasing model parameters actually hurt network performance, so future studies will be aimed at investigating the relationship between the number of model parameters and the performance of the network. Additionally, it will be important to determine the interaction between specific network implementations and the ideal Activation Function wrapping for slope-determining neurons. Lastly, it may be useful to investigate layer-wide single-node modulation on models with parallel LSTM's.

Epigenetics refers to the activation and inactivation of genes (Weinhold, 2006), often as a result of environmental factors. These changes in gene-expression result in modifications to the generation and regulation of cellular proteins, such as ion channels, that regulate how the cell controls the flow of current through the cell membrane (Meadows et al., 2016). The modulation of these proteins will strongly influence the tendency of a neuron to fire and hence affect the neurons function as a single computational node. These proteins, in turn, can influence epigenetic expression in the form of dynamic control (Kawasaki et al., 2004).

Regarding the effects of these signals, we can compare the output of neurons and nodes from a variety of perspectives. First and foremost, intrinsic excitability refers to the ease with which a neurons electrical potential can increase, and this feature has been found to impact plasticity itself (Desai et al., 1999). From this view, the output of a node in an artificial neural network would correspond to a neurons firing rate, which Intrinsic Excitability is a large contributor to, and our extra gate would be setting the node's intrinsic excitability. With the analogy of firing rate, another phenomenon can be considered. Neurons may experience various modes of information integration, typically labeled Type 1 and Type 2. Type 1 refers to continuous firing rate integration, while Type 2 refers to discontinuous information (Tateno et al., 2004). This is computationally explained as a function of interneuron communication resulting in neuron-activity nullclines with either heavy overlap or discontinuous saddle points (Miller, 2016). In biology, a neuron may switch between Type 1 and Type 2 depending on the presence of neuromodulator (Stiefel & Gutkin, 2012). Controlling the degree to which the *tanh* function encodes to a binary space, our modification may be conceived as determining the form of information integration. The final possible firing rate equivalence refers to the ability of real neurons to switch between different firing modes. While the common mode of firing, Tonic firing, generally encodes information in rate frequency, neurons in a Bursting mode (though there are many types of bursts) tend to encode information in a binary mode - either firing

bursts or not (Tateno et al., 2004). Here too, our modification encompasses a biological phenomenon by enabling the switch between binary and continuous information.

Another analogy to an ANN nodes output would be the neurotransmitter released. With this view, our modification is best expressed as an analogy to Activity Dependent Facilitation and Depression, phenomena which cause neurons to release either more or less neurotransmitter. Facilitation and depression occur in response to the same input: past activity (Reyes et al., 1998). Our modification enables a network to use previous activity to determine its current sensitivity to input, allowing for both Facilitation and Depression. On the topic of neurotransmitter release, neuromodulation is the most relevant topic to the previously shown experiments. Once again, Marblestone et al. (2016) explains the situation perfectly, expressing that research (Bargmann, 2012; Bargmann & Marder, 2013) has shown "the same neuron or circuit can exhibit different input-output responses depending on a global circuit state, as reflected by the concentrations of various neuromodulators". Relating to our modification, the slope of the activation function may be conceptualized as the mechanism of neuromodulation, with the new gate acting analogously to a source of neuromodulator for all nodes in the network.

Returning to a Machine Learning approach, the ability to adjust the slope of an Activation Function has an immediate benefit in making the back-propagation gradient dynamic. For example, for Activations near 0, where the *tanh* Function gradient is largest, the effect of our modification on node output is minimal. However, at this point, our modification has the ability to decrease the gradient, perhaps acting as pseudo-learning-rate. On the other hand, at activations near 1 and -1, where the *tanh* Function gradient reaches 0, our modification causes the gradient to reappear, allowing for information to be extracted from inputs outside of the standard range. Additionally, by implementing a slope that is conditional on node input, the node has the ability to generate a wide range of functional Activation Functions, including asymmetric functions. Lastly, injecting noise has been found to help deep neural networks with noisy datasets (Zheng et al., 2016), which is noteworthy since noise may act as a stabilizer for neuronal firing rates, (Touboul et al., 2012). With this in mind, Table 3.2 demonstrates increased clustering in two-dimensional node-Activation space, when the Activation Function slope is made to be dynamic. This indicates that noise may be a mediator of our modification, improving network performance through stabilization, induced by increasing the variability of the input-output relationship.

In summary, we have shown evidence that nodes in LSTMs and CNNs benefit from added complexity to their input-output dynamic. Specifically, having a node that adjusts the slope of the main layer's nodes' activation functions mimics the functionality of neuromodulators and is shown to benefit the network. The exact mechanism by which this modification improves network performance remains unknown, yet it is possible to support this approach from both a neuroscientific and machine-learning perspective. We believe this demonstrates the need for further research into discovering novel non-computationally-demanding methods of applying principles of neuroscience to artificial networks.

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

# 6 APPENDIX

## 6.1 SUPPLEMENTARY DATA METHODOLOGY

Additionally we tested our modulator gate, with $\tau_l(\cdot)$ set to *sigmoid*, on a much more computationally demanding three-layered LSTM network with weight drop method named *awd-lstm-lm* (Merity et al., 2017; 2018). This model was equipped to handle the Penn-Treebank dataset (Marcus et al., 1993) and was trained to minimize word perplexity. The network was trained for 500 epochs, however, the sample size was limited due to extremely long training times.

Table 6: Network parameters per cell

| Model | Modulated | Control |
|-------|-----------|---------|
| awd-lstm-lm | 6.61 M | 5.29 M |

## 6.2 SUPPLEMENTARY DATA RESULTS

On the Penn-Treebank dataset with the *awd-lstm-lm* implementation, sample size was restricted to 2 per condition, due to long training times and limited resources. However on the data collected, our model outperformed template perplexity, achieving an average of 58.4730 compared to the template average 58.7115. Due to the lack of a control for model parameters, interpretation of these results rests on the assumption that the author fine-tuned network parameters such that the template parameters maximized performance.

# 7 SUPPLEMENTARY DATA FIGURES & TABLES

## 7.1 AWD-LSTM-LM ON PENN-TREEBANK

Table 7: Comparison of mean test Perplexities *lower = better*

| Model | Epochs | Modulated | Control | Statistical Analysis |
|-------|--------|-----------|---------|----------------------|
| awd-lstm-lm on Penn-Treebank | 500 | **58.4730** | 58.7115 | T: 1.842
DOF: 1.9
Hedges's G: 1.853 |

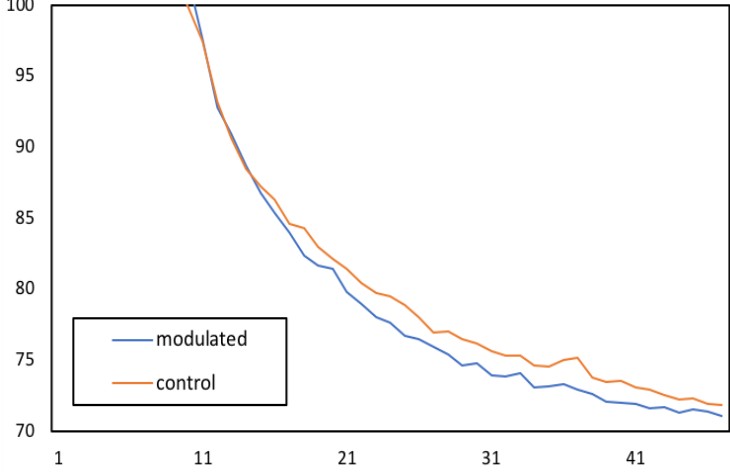

Figure 7: Validation Perplexity progress *(lower = better)*

## 7.2 SUPPLEMENTAL LSTM DATA

Table 8: LSTM test performance histograms illustrating improved test-retest reliability

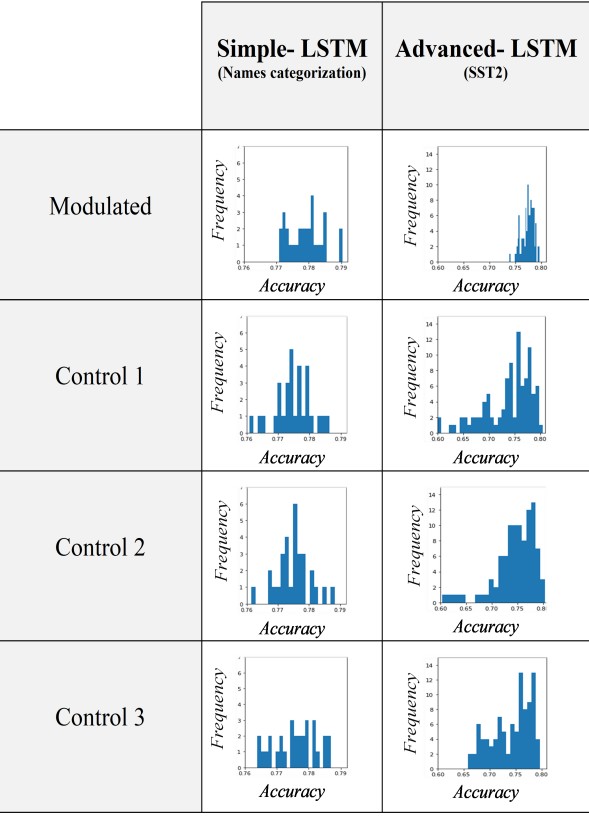

