# OpenReview forum: "Context Dependent Modulation of Activation Function"
_ICLR.cc/2019/Conference_

### Official Review · AnonReviewer2 · 2018-11-03
**modulating scalar applied per-neuron**

**Rating:** 6
**Confidence:** 4

**Review:**

The paper introduces a new twist to the activation of a particular neuron. They use a modulator which looks at the input and performs a matrix multiplication to produce a vector. That vector is then used to scale the original input before passing it through an activation function. Since this modulating scalar can look across neurons to apply a per-neuron scalar, it overcomes the problem that otherwise neurons cannot incorporate their relative activation within a layer. They apply this new addition to several different kinds of neural network architectures and several different applications and show that it can achieve better performance than some models with more parameters.


Strengths:
- This is a simple, easy-to-implement idea that could easily be incorporated into existing models and frameworks.
- As the authors state, adding more width to a vanilla layer stops increasing performance at a certain point. Adding more complex connections to a given layer, like this, is a good way forward to increase capacity of layers.
- They achieve better performance than existing baselines in a wide variety of applications.
- The reasons this should perform better are intuitive and the introduction is well written.

Weaknesses:
- After identifying the problem with just summing inputs to a neuron, they evaluate the modulator value by just summing inputs in a layer. So while doing it twice computes a more complicated function, it is still a fundamentally simple computation.
- It is not clear from reading this whether the modulator weights are tied to the normal layer weights or not. The modulator nets have more parameters than their counterparts, so they would have to be separate, I imagine.
- The authors repeatedly emphasize that this is incorporating "run-time" information into the activation. This is true only in the sense that feedforward nets compute their output from their input, by definition at run-time. This information is no different from the tradition input to a network in any other regard, though.
- The p-values in the experiment section add no value to the conclusions drawn there and are not convincing.

Suggested Revisions:
- In the abstract: "A biological neuron change[s]"
- The conclusion is too long and adds little to the paper

---

> ### Author Response · Authors · 2018-11-24
> **Thank you very much for your helpful comments!**
>
> I.1 We used a modulation to change the shape of the activation, it is still a single computation but the effect is multiplicative instead of additive and effects an entire layer rather than a single node.
> I.2 Our modulation was set on the convolution layer only before the activation layer. It adds a very small amount of parameters, we also did a ‘lite’ DenseNet version to compare our modification with fewer parameters.
> I.3 We used ‘run-time’ in the manuscript trying to express that our modulation method can take the input and use that information and change the shape of the activation function on the fly.
> II. we thank the reviewer for picking out minor errors, all suggestions taken.

---

### Official Review · AnonReviewer3 · 2018-11-04
**Interesting, but no convincing results and analysis**

**Rating:** 4
**Confidence:** 5

**Review:**

This paper proposes a scalar modulator adding to hidden nodes before an activation function. The authors claim that it controls the sensitivity of the hidden nodes by changing the slope of activation function. The modulator is combined with a simple CNN, DesneNet, and a LSTM model, and they provided the performance improvement over the classic models.

The paper is clear and easy to understand. The idea is interesting. However, the experimental results are not enough and convincing to justify it.

1) The authors cited the relevant literature, but there is no comparison with any of these related works.

2) Does this modulator actually help for CNN and LSTM architectures? and How? Recently, there are many advanced CNN and LSTM architectures. The experiments the authors showed were with only 2 layer CNNs and 1 layer LSTM. There should be at least some comparison with an architecture that contains more layers/units and performs well. There is a DenseNet comparison, but it seems to have an error. See 4) for more details.

3) The authors mentioned that the modulator can be used as a complement to the attention and gate mechanisms. Indeed, they are very similar. However, the benefit is unclear. More experiments need to be demonstrated among the models with the proposed modulator, attention, and gates, especially learning behavior and performance differences.

4) The comparison in Table 2 is not convincing.
- The baseline is too simple. For instance on CIFAR10, a simple CNN architecture introduced much earlier (like LeNet5 or AlexNet) performs better than Vanilla CNNs or modulated CNNs.
- DenseNet accuracy reported in Table 2 is different from to the original paper: DenseNet (Huang et al. 2017) CIFAR10 # parameters 1.0M, accuracy 93%, but in this paper 88.9%. Even the accuracy of modulated DenseNet is 90.2% which is still far from the original DenseNet.
Furthermore, there are many variations of DenseNet recently e.g., SparsenNet: sparsified DenseNet with attention layer (Liu et al. 2018), # parameters 0.86M, accuracy 95.75%. Authors should check their experiments and related papers more carefully.

Side note: page 4, Section 3.1 "The vanilla DenseNet used the structure (40 in depth and 12 in growth-rate) reported in the original DenseNet paper (Iandola et al., 2014)". This DenseNet structure is from Huang et al. 2017 not from Iandola et al. 2014.

---

> ### Author Response · Authors · 2018-11-24
> **Thank you very much for your helpful comments!**
>
> 1. We did not compare performances with relevant works since we are generally not trying to compete against the existing tools (i.e. attention). Our modification focused on a different aspect of changing the slope of the activation and can be applied on top of the related works. We will explore other implementations of existing works in conjunction with modulator nodes in the future works.
> 2. Regarding the CNN model performance, in the paper, we didn’t report the best DenseNet as the baseline from the original work. After we tried more complexed model setup, we got the following new results:
>
> Model				                               Top-1 Accuracy
> ----------------------------------------------------------------------------
> DenseNet-161(k=48)		                        93.79
> ModulatedDenseNet-161(k=48)		93.95
>
> We also thank the reviewer for mentioning other recent works, we will explore the comparison in the future works.

---

### Official Review · AnonReviewer1 · 2018-11-07
**Idea with no very convincing benefits, baseline comparison to improve.**

**Rating:** 4
**Confidence:** 4

**Review:**

Summary: this submission proposes a modification of neural network architectures that allows the modulation of activation functions of a given layer as a function of the activations in the previous layer. The author provide different version of their approach adapted to CNN, DenseNets and LSTM, and show it outperforms a vanilla version of these algorithms.
Evaluation: In the classical context of supervised learning tasks investigated in this submission, it is unclear to me what could be the benefit of introducing such “modulators”, as vanilla ANNs already have the capability of modulating the excitability of their neurons. Although the results show significant, but quite limited, improvements with respect to the chosen baseline, more extensive baseline comparisons are needed.

Details comments:
1.	Underlying principles of the approach
It is unclear to me why the proposed approach should bring a significant improvement to the existing architectures. First, from a neuroscientific perspective, neuromodulators allow the brain to go through different states, including arousal, sleep, and different levels of stress. While it is relatively clear that state modulation has some benefits to a living system, it is less so for an ANN focused on a supervised learning task. Why should the state change instead of focusing on the optimal way to perform the task? If the authors want to use a neuroscientific argument, I would suggest to elaborate based on the precise context of the tasks they propose to solve.
In addition, as mentioned several times in the paper, neuromodulation is frequently associated to changes in cell excitability. While excitability is a concept that can be associated to multiple mechanisms, a simple way to model changes in excitability is to modify the threshold that must be reached by the membrane potential of a given neuron in order for the cell to fire. Such simple change in excitability can be easily implemented in ANNs architectures by affecting one afferent neuron in the previous layer to the modification of this firing threshold (simply adding a bias term). As a consequence, if there is any benefit to the proposed architecture, it is very likely to originate specifically from the multiplicative interactions used to implement modulation in this paper. However, approximation of such multiplicative interactions can also be implemented using multiple layers network equipped with non-linear activations. Overall, it would be good to discuss these aspects in great detail in the introduction and/or discussion of the paper, and possibly find a more convincing justification for the approach.

2.	Weak baseline comparison results
In the CNN experiments, modulated networks are only compared with a single vanilla counterpart equipped with ReLu. There are at least two obvious additional baseline comparison that would be useful: what if the Re-Lu activations are replaced with fixed sigmoids? And what if batch-normalization is switched on/off (I could not find whether it was used at all). Indeed it, we should exclude benefits that are simply due to the non-linearity of the sigmoid, and batch normalization also implements a form of modulation at training that may provide benefits equivalent to modulation (or on the contrary, batch norm could implement a modulation in the wrong way). It would be better to look at all possible combinations of these architecture choices.
Due to lack of details in the paper and my personal lack of expertise in LSTMs, I will not comment on baselines for that part but I assume similar modifications can be done.
Overall, given the weak improvements in performance, it is questionable whether this extra degree of complexity should be added to the architecture. Additionally, I could not find the precise description of the statistical tests performed. Ideally, the test, the number of samples, the exact p-value, and whether the method of correction for multiple comparison should be included each time a p-value is mentioned.

---

> ### Author Response · Authors · 2018-11-24
> **Thank you very much for your helpful comments!**
>
> 1. The neuroscientific inspiration came simply from looking at what neurons were capable of, as opposed to a descriptive approach for why their capabilities may be useful for specific tasks. However, why would modulation benefit a supervised learning task is a completely valid and absolutely vital question. We did not find an easy location to address this question in the paper. However, for a brief explanation, supervised learning may benefit from the increased contrast in the amplitude of the signals propagating through the network. A sigmoidal modulator should theoretically learn to spatiotemporally inhibit signals, thereby increasing the relative gain of certain signals or pathways based. This context modulation is common in the visual system via reciprocal inhibition. Regarding ‘Intrinsic Excitability’, it is very true that a bias can capture it if node activation is envisioned as subthreshold voltage, however, if node activation is considered analogous to firing rate, once a neuron passes a threshold,  ‘Intrinsic Excitability’ has a multiplicative effect on the firing rate frequency.
> 2. We thank the reviewer for suggesting the possible experimental setups that not included in this paper. We will strengthen the experiment section in the final paper and explore more in future works.

---

### Official Review · AnonReviewer5 · 2018-11-09
**Interesting idea but the overall state of the paper needs improvements**

**Rating:** 4
**Confidence:** 3

**Review:**

Summary:
This paper introduces an architectural change for basic neurons in neural network. Assuming a "neuron" consists of a linear combination of the input, followed by a non-linear activation function, the idea is to multiply the output of the linear combination by a "modulator", prior to feeding it into the activation function. The modulator is itself a non-linear function of the input. Furthermore, in the paper's implementation, the modulators share weights across the same layer. The idea is demonstrated on basic vision and NLP tasks, showing improvements over the baselines.

I - On the substance:
1. Related concepts and biological inspirations
The idea is analogous to attention and gating mechanisms, as the authors point out, with the clear distinction that the modulation happens _before_ the activation function. It would have been interesting to experiment a combination of modulation and attention since they do not act on the same levels.
Also, the authors claim inspiration from the biological neurons, however, they do not elaborate in depth on the connections to the neuronal concepts mentioned in the introduction.

2. The performance of the proposed approach
In the first experiment, the modulated CNN at 150 epochs seems to have comparable performance with the vanilla CNN at 60 (the latter CNN starts overfitting afterwards). Why not extending the learning curve to more epochs since the modulated CNN seems on a positive slope?
The other experiments show some improvements over the baselines, however more experiments are necessary for claiming generality. Especially, the baselines remain too simple and there are some well-known well-performing architectures, for both image and text processing, that the authors could compare to (cf winning architectures for imagenet for instance). They could also take these same architectures and augment them with the modulation proposed in the paper.
Furthermore, an ablation study is clearly missing, what about different activation functions, combination with other optimization techniques etc.?

II - On the form:
1. the paper is sometimes unclear, even though the overall narrative is sound,
2. wiggly red lines are still present in the caption of Figure 1 right.
3. Figure 6 could be greatly simplified by putting its content in the form of a table, I don't find that the rectangles and forms bring much benefit here.
4. Table 5 (should it not be Figure?): it is not fully clear what the lines represent and based on which input.
5. some typos:
 - abstract: a biological neuron change[s]
 - abstract: accordingly to -> according to
 - introduction > paragraph 2 > line 11: Each target node multipl[i]es

III - Conclusion:
The idea is interesting and some of the experiments show nice results (eg. modulated densenet-lite outperforming densenet) but the overall paper needs further improvements. In particular, the writing needs to be reworked, the experiments to be consolidated, and the link to neuronal modulation to be further investigated.

---

> ### Author Response · Authors · 2018-11-24
> **Thank you very much for your helpful comments!**
>
> I.1.  We thank the reviewer for pointing out the comparison of our modification with the attention mechanism. Since we stated in the paper that our modification focused on a different aspect which is the slope of the activation function, we didn’t include this kind of comparison. In future works, we will explore the possibility of combining both methods.
> I.2.  From our experiments, the modulated vanilla network structure can outperform the counterpart by a small margin. After the epochs showed in the chart, the performance of the modulated network became almost flat. We tried to set the type of activation functions and optimization methods the same for comparing the performance of models in our work. Also, we can explore more combinations of setups in future works.
> II. we thank the reviewer for picking out minor errors, all suggestions taken.

---

### Official Review · AnonReviewer4 · 2018-11-13
**Restricted/simplified version of network in network by Lin et. al. without clear benefits**

**Rating:** 4
**Confidence:** 5

**Review:**

Paper summary:

This paper proposes a method to scale the activations of a layer of neurons in an ANN depending on the inputs to that layer. The scaling factor, called modulation, is computed using a separate weight matrix and activation function. It is multiplied with each neuron's activation before applying its non-linearity. The weight matrix of the modulator is learned alongside the other weights of the network by backpropagation. The authors evaluate this modulated neural unit in convolutional neural networks, densely connected CNNs and recurrent networks consisting of LSTM units. Reported improvements above the baselines are between 1% - 3%.

Pro:

+ With some minor exceptions the paper is clearly written and comprehensible.
+ Experiments seem to have been performed with due diligence.
+ The proposed modulator is easy to implement and applicable to (almost) all network architectures.

Contra:

- Lin et. al. (2014) proposed a network in network architecture. In this architecture the output of each neural unit is computed using a small neural network contained in it and thus arbitrary, input-dependent activation functions can be realized and learned by each neuron. The proposed neural modulation mechanism in the paper at hand is in fact a more restricted version of the network-in-network model and the authors should discuss the relationship of their proposal to this prior work.

- When comparing the test accuracy of CNNs in Fig. 4 the result is questionable. If training of the vanilla CNN was stopped at its best validation loss (early stopping), the difference in accuracies would have been marginal. Also the choice of hyper-parameters may significantly affect the outcome of the comparison experiments. More experiments would be necessary to prove the advantage of this model over a wide range of hyper-parameters.

Minor points:

- It is unclear whether the modulator weights are shared along the depth of a CNN layer, i.e. between feature maps.

- Page 9: "Our modification enables a network to use previous activity to determine its current sensitivity to input [...]" => A vanilla LSTM is already capable of doing that using its input gate.

- Page 9: "[...] the ability to adjust the slope of an Activation Function has an immediate benefit in making the back-propagation gradient dynamic." => In fact ReLUs do not suffer from the vanishing gradient problem. Furthermore DenseNets already provide a short-path for the gradient flow by introducing skip connections.

- The discussion at the end adds little value and rather seems to be a motivation of the model than a discussion of the results.

Rating:

My main concern is that the proposed modulator is a version of the network in network model restricted to providing a scaling factor. Although the authors motivate this model biologically, I do not see sufficient empirical evidence to believe that it is advantageous over the full network in network model by Lin et. al. I would recommend to add a direct comparison to that model to a future version of this paper.

---

> ### Author Response · Authors · 2018-11-21
> **Thank you for your helpful comments!**
>
> 1. We thank the reviewer for pointing out the ‘Network in Network’ paper (Lin et. al. 2014), we will add the discussion with this work in our final version. In short, our approach applied the context modulation only before the activation layer, on the contrary, Lin et. al.’s method was applied to every convolutional layer; also, the modulator weights were applied to all the feature maps providing a very easy to implement light weighted modification that was solely used to change the activation function slope.
> 2. In vanilla LSTM, the input gate can control how much the input will affect the cell status, but our modification focuses on a different part which is adding a modulator to control the shape of the activation function.
> 3. We focus on the context dynamic activation function which can have a side benefit of easing the gradient issue of other activation functions e.g. tanh in LSTM.
> 3. And we will clear the discussion section to make our claim more clear.

---

### Meta-Review · Area_Chair1 · 2018-12-12
**the merit needs to be validated**

**Confidence:** 5
**Recommendation:** Reject

**Metareview:**

The paper adds a new level of complexity to neural networks, by modulating activation functions of a layer as a function of the previous layer activations.  The method is evaluated on relatively simple vision and language tasks.

The idea is nice, but seems to be a special case of previously published work; and the results are not convincing.  Four of five reviewers agree that the work would benefit from: improving comparisons with existing approaches, but also improving its theoretical framework, in light of competing approaches.